# Systematic mediation and interaction analyses of kidney function genetic loci in a general population study

Dariush Ghasemi-Semeskandeh[1,2,3*], David Emmert[2], Eva König[2], Luisa Foco[2], Martin Gögele[2], Mohamed S. Sarhan[2,4], Laura Barin[2], Ryosuke Fujii[2,5,6], Christian Fuchsberger[2], Dorien J. M. Peters[1], Peter P. Pramstaller[2], Cristian Pattaro[2*]

**1** Department of Human Genetics, Leiden University Medical Center, Leiden, The Netherlands, **2** Institute for Biomedicine, Eurac Research, Bolzano, Italy, **3** Integrative Data Analysis Unit, Health Data Science Centre, Human Technopole, Milan, Italy, **4** Department CIBIO, University of Trento, Trento, Italy, **5** Department of Preventive Medical Sciences, Fujita Health University School of Medical Sciences, Toyoake, Japan, **6** Department of Preventive Medicine, Nagoya University Graduate School of Medicine, Nagoya, Japan

* ghasemi.dariush@yahoo.com (DG-S); cristian.pattaro@eurac.edu (CP)

## Abstract

Chronic kidney disease (CKD) is a complex disease affecting >10% of the global population, with large between- and within-continent variability reflecting major environmental determinants. To identify molecular targets for treatment and prevention, genome-wide association study meta-analyses (GWAMAs) of CKD-defining traits have identified hundreds of genetic loci in aggregated population samples. However, while GWAMAs estimate the average allelic effect across studies, single population studies may be relevant to unravel specific mechanisms. To assess whether a study sample from a specific population could extend existing knowledge on kidney function genetics, we selected 147 kidney function relevant loci identified by a large GWAMA, assessing their association with the glomerular filtration rate estimated from serum creatinine (eGFRcrea) in 10,146 participants to the Cooperative Health Research In South Tyrol (CHRIS) study, conducted in an Alpine region where thyroid dysfunction is common. We identified associations with single nucleotide polymorphisms (SNPs) at 11 loci, showing up-to-5.4 times larger effect sizes than in the corresponding GWAMA, not explainable by allele frequency differences. Systematic mediation analysis across 70 quantitative traits identified serum magnesium and the activated partial thromboplastin time as partial mediators of the eGFRcrea associations at *SHROOM3* and *SLC34A1*, respectively. Given that free triiodothyronine and thyroxine acted as effect modifiers across all loci, we conducted SNP-by-thyroid stimulating hormone (TSH) interaction analyses, identifying significant interactions at *STC1*: SNPs had larger effects on eGFRcrea at higher TSH levels, possibly reflecting stanniocalcin-1 autocrine and paracrine role. Individual population studies can help characterize genetic associations. The interplay between phenotypes at *SHROOM3*

**Data availability statement:** The data used in the current analyses can be requested with an application to the CHRIS Access Committee at access.request.biomedicine@eurac.edu.

**Funding:** The CHRIS Study was funded by the Autonomous Province of Bolzano-South Tyrol Department of Innovation, Research, University and Museums and supported by the European Regional Development Fund (FESR1157). This work was carried out within the TrainCKDis project, funded by the European Union's Horizon 2020 research and innovation programme under the Marie Skłodowska-Curie grant agreement H2020-MSCA-ITN-2019 ID:860977 (TrainCKDis).

**Competing interests:** The authors have declared that no competing interests exist.

and *SLC34A1* and the role of thyroid function as a genetic effect modifier warrant further investigations.

## Introduction

Chronic kidney disease (CKD) is a common complex disease that increases the risk of kidney failure, cardiovascular mortality, and all-cause mortality [1]. CKD is predicted to become the 5th leading cause of death by 2040 [2]. While affecting >10% of the population globally, CKD prevalence shows marked variability across countries, both between and within continents [3]. Between-continent variability is likely guided by combinations of environmental and genetic differences, such as is the case with sickle cell trait and albuminuria [4]. Within-continent variability could be more related to environmental differences, including lifestyle, public health policies, and assessment methods [4,5].

To unravel the genetic basis of CKD, several genome-wide association study meta-analyses (GWAMAs) were conducted that analyzed the CKD-defining marker glomerular filtration rate (GFR) estimated from serum creatinine (eGFRcrea) [6]. More than 1000 associated loci have been identified to date [7]. While GWAMAs estimate the average allelic effect across studies, evidence shows that individual population studies may be extremely relevant to advance discoveries not only in the field of rare diseases [8] but also with regard to markers of common chronic diseases such as triglycerides and low density lipoprotein (LDL) cholesterol [9], age at diabetes onset [10], and mood disorders [11]. The evidence accumulated so far suggests that there is no general rule and that the contribution of an individual study to the dissection of common phenotypes' biological basis depends on study-specific characteristics and, as such, it can only be assessed on a case-by-case basis.

Here we analyzed genetic associations with eGFRcrea in the Cooperative Health Research In South Tyrol (CHRIS) study, a general population study conducted in a mountainous region of central Europe [12]. A sample from this same population was previously instrumental to identify genetic linkage of serum creatinine with the *APOL1-MYH9* locus [13] and to replicate common variants associated with eGFRcrea [14]. Characteristics of the geographical region where the study was conducted include the demographic stability over time [15], the rural environment, and a high frequency of hypothyroidism, typical of mountainous regions [16]. Aim of our analysis was to assess whether this particular study sample could provide information that extend current knowledge on kidney function genetics, by leveraging on local specific characteristics.

We focused our investigation on 147 loci identified by a large GWAMA of the CKD-Gen Consortium with proven kidney function relevance [17]. In the cited work, the CKD-Gen identified 308 loci associated with eGFRcrea at genome-wide significance level; of them, 264 were replicated in an independent population study and thus validated; but given eGFRcrea might also reflect creatinine metabolism in addition to kidney function, the authors further filtered their results for direction-consistent association with blood urea nitrogen (BUN), an alternative marker of kidney function [17]. These loci were

tested for association with eGFRcrea in the CHRIS study. The purpose of this analysis was to identify those loci for which a study of limited size like CHRIS could try to contribute new information. We conveniently applied a multiple-testing penalty that subsets 11 loci with most convincing relevance for the CHRIS study. The features of such 11 loci were then assessed against corresponding European ancestry CKDGen GWAMA results. Given that several of these loci exhibited larger effects in CHRIS than in the European ancestry GWAMA, we examined the possible reasons of such larger effects by conducting extensive mediation analyses across 70 quantitative biochemical and anthropometric traits, reflecting multiple health conditions. Further, after observing a systematic modification of the genetic association with eGFRcrea when adjusting for thyroid-related traits, we conducted SNP-by-thyroid function interaction analyses. The aims of the mediation and the interaction analyses were to identify the presence of intermediate phenotypes and effect modifiers, respectively, that were specific of the study sample.

## Methods

### Study sample

The present analyses were based on 10,146 individuals with complete genotype data who participated to the CHRIS study, a population-based study conducted in South Tyrol, Italy, between 2011 and 2018, which was approved by the Ethics Committee of the Healthcare System of the Autonomous Province of Bolzano-South Tyrol, protocol no. 21/2011 (19 Apr 2011). All participants provided written informed consent.

CHRIS has been extensively described elsewhere [12,18]. Briefly, participants underwent blood drawing, urine collection, anthropometric measurements, and clinical assessments early in the morning, after an overnight fast. Medical history was reconstructed through interviewer- and self-administered standardized questionnaires. Drug treatment was identified via barcode scan of the drug containers that participants were instructed to bring at the study center visit, with the drug code linked to an official drug databank [12].

### eGFRcrea estimation

Serum creatinine (SCr) was measured with a colorimetric assay on Roche Modular PPE (n = 4,176) and Abbott Diagnostic Architect c16000 (n = 5,970) instrumentations. According to previous analyses [18], SCr was normalized by instrument (fixed effect) and participation period (random effect) using a linear mixed model implemented in the 'lme4' R package [19]. To reflect the same methods used in the CKDGen Consortium meta-analysis [17], eGFRcrea was estimated using the 2009 CKD-EPI equation [20] implemented in the R package 'nephro' v1.2 [21] and winsorized at 15 and 200 ml/min/1.73m$^2$. Finally, the natural logarithm transformation was applied.

### Genotype imputation

Genotyping was performed in two batches using genotyping array chips based on the Illumina Human Omni Microarray platform. Following quality control analysis, the two data batches were merged, phased with SHAPEIT2 v2.r837, using the duoHMM method (--duohmm -W 5) with 800 states and 30 rounds [22], and imputed based on the TOPMed r2 standard reference panel on the Michigan Imputation Server [23]. We obtained 34,084,280 SNPs with minimum imputation quality index Rsq ≥ 0.3 and minor allele count (MAC) ≥1, aligned with human genome assembly GRCh38. A genetic relatedness matrix (GRM) was obtained from the autosomal genotyped SNPs using EPACTS v3.2.6 modified for compatibility with assembly GRCh38. Genetic principal components (PCs) were estimated based on the genotyped variants data with minor allele frequency (MAF) >0.05 using GCTA [24].

### Genetic principal component analysis (PCA)

We performed PCA on autosomal variants from the 1000 Genomes Project Phase 3 (1000GPh3) and the CHRIS study to assess population structure. 1000GPh3 VCF files for chromosomes 1–22 were downloaded, and a list of target variant

sites was extracted from the CHRIS study genotype dataset. Variants from the 1000GPh3 dataset matching these sites were extracted using vcftools, and chromosome-specific VCF files were merged into a single dataset. To obtain a set of independent SNPs, linkage disequilibrium (LD) pruning was conducted using PLINK (--indep-pairwise 100kb 5 0.1), retaining variants with r²<0.1 within 100-kb windows. PCA was conducted on the pruned dataset, extracting per-SNP weights for the first 20 PCs. Individuals from both datasets were projected onto the PC space using PLINK's --score function, generating scores for each individual for the first four PCs.

### Identification of locally relevant loci

We conducted a GWAS of age- and sex-adjusted residuals of ln(eGFRcrea) using the EMMAX method [25] implemented in EPACTS v3.2.6, including the GRM to model the sample structure. After the analysis, we removed variants with minor allele frequency (MAF) <0.005, leaving 10,158,100 SNPs for further characterization. Genomic inflation was assessed on this set of SNPs by estimating the genomic control factor λ [26].

Summary statistics from the CKDGen GWAMA [17] were downloaded from the publicly available repository at https://ckdgen.imbi.uni-freiburg.de. Given a subset of 4661 samples from the CHRIS study was previously included in such GWAMA, to ensure sample independence we removed CHRIS data from the GWAMA summary statistics using Meta-Subtract v1.60 [27]. Removing CHRIS did not alter the CKDGen results of the 147 variants. We lifted the CKDGen genomic positions from the GRCh37 to the GRCh38 map, using CrossMap v0.5.3 [28].

Given small-sample bias can cause large effect estimate fluctuations and given it is not uncommon to observe small studies with opposite effect directions compared to large GWAMA results, to ensure consistency of the effect direction, the association testing of the 147 loci in CHRIS was based on a one-sided test, following common practice [29]. Statistical significance was set at the Bonferroni-corrected level of 0.00034 (0.05/147), for any variant in strong LD (r²>0.8) with a lead CKDGen variant. LD was estimated using emeraLD v0.1 [30]. The total number of SNPs within the 147 loci was 6337. Adjustment for 147 independent tests covered 98% of the SNP's cumulative variance (S1 Fig in S1 File).

We compared the variance of ln(eGFRcrea) explained by SNPs in CHRIS versus CKDGen European ancestry analysis. To allow comparison, we used the approximate formulas $b_{CKDGen}^2 \times 2p_{CKDGen}(1-p_{CKDGen})/var(y)$ versus $b_{CHRIS}^2 \times 2p_{CHRIS}(1-p_{CHRIS})/var(y)$ [17], where b is the estimated SNP effect, p is the SNP allele frequency, and var(y) is the variance of the age- and sex-adjusted ln(eGFRcrea) in CHRIS.

### Mediation analysis

The ln(eGFRcrea)-SNP associations identified in the CHRIS study were submitted to mediation analysis across 70 quantitative anthropometric, blood pressure and biochemical traits (S1 Table in S2 File), following the flowchart represented in Fig 1. SCr was included as a positive control. To prevent potential measurement instrument effects [18], we applied quantile normalization to each trait (Supplementary Methods in S1 File). Mediation analysis was conducted for each *trait* following the established 4-step framework and fitting linear regression models throughout:

1. ln(eGFRcrea)= $\beta_1$ SNP +$\sum_i^K \theta_i X_i$+ $\varepsilon_1$

2. ln(eGFRcrea)= $\beta_2$ SNP + $\gamma_2$ *trait* + $\sum_i^K \theta_i X_i$ + $\varepsilon_2$

3. *trait*= $\beta_3$ SNP + $\sum_i^K \theta_i X_i$ + $\varepsilon_3$

4. *trait*= $\beta_4$ SNP + $\delta_4$ ln(eGFRcrea) + $\sum_i^K \theta_i X_i$ + $\varepsilon_4$

where $\varepsilon_1$ to $\varepsilon_4$ refer to Gaussian error terms, and $\sum_i^K \theta_i X_i$ indicate the inclusion of K covariates, namely age, sex, the first 10 genetic PCs and an intercept. Fitting PC-adjusted simple linear models came for practical reasons after observing substantial equivalence with kinship-adjusted linear mixed models obtained with EMMAX (S2 Fig in S1 File). Step 1 corresponds to the GWAS results, with $\beta_1$ ($b_1$) being always significant given the selection process.

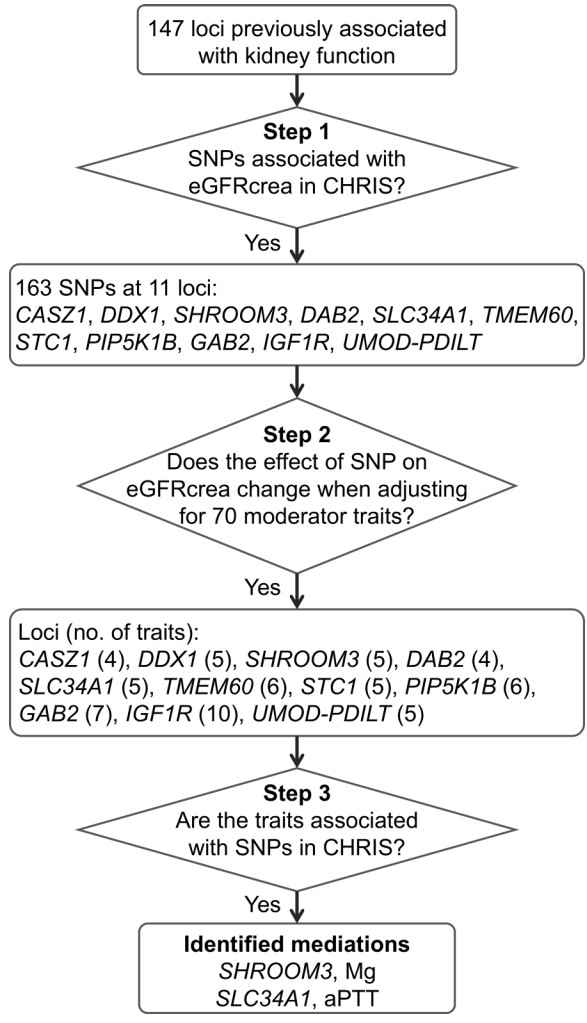

**Fig 1. Flowchart of the SNP and trait selection leading to mediation analysis.** Abbreviations: Mg, serum magnesium; aPTT, activated partial thromboplastin time.

We defined two criteria that must be satisfied in order for a trait to be considered at least a partial mediator of the ln(eGFRcrea)-SNP relation: (i) evidence of a substantial alteration of the SNP effect on ln(eGFRcrea) when adjusting for the trait (step 2) and (ii) the SNP is also associated with the trait (step 3) at a multiple-testing corrected level of $6.5 \times 10^{-5}$ (corresponding to 0.05/ (70 traits × 11 independent loci)). Criterion (i) was assessed by analyzing the distribution of the estimate of $\beta_2$ ($b_2$): for a specific trait, $b_2$ was classified as an outlier according to the rule $b_2 < P_{10} -$ 1.5 $(P_{75} - P_{25})$ or $b_2 > P_{90} + 1.5 (P_{75} - P_{25})$, where $P_{10}$, $P_{25}$, $P_{75}$, and $P_{90}$ indicate the 10th, 25th, 75th, and 90th percentile of the distribution of $b_2$ across all traits. This is essentially a more stringent version of the Tukey's rule for outlier detection.

To support associations in step 3 and assess whether lack of mediation evidence might have been due to lack of power in the CHRIS study, we interrogated the SNP-trait associations that passed the mediation analysis step 2 on the PhenoScanner v2 [31] (interrogated on 31-Jan-2023), GWAS Catalog [32] (https://www.ebi.ac.uk/gwas/docs/api; 24-Feb-2023), and the ThyroidOmics Consortium summary statistics [33] (https://www.thyroidomics.com), for associations at $P < 5.0 \times 10^{-8}$.

We queried variants for which we observed potentially mediated effects in the European ancestry datasets of the GTEx Consortium v10 database (https://gtexportal.org/home/; 1-Mar-2025) across 49 tissues (n = 65–573 samples per tissue) using GTEx API v2 and in the Human Kidney eQTL data [34] (n = 686), to assess association with gene expression (eQTL) at $P < 5 \times 10^{-8}$. To identify protein quantitative trait loci (pQTLs) at $P < 5 \times 10^{-8}$, we interrogated GWAS summary results for 2923 plasma proteins on 54,219 UK Biobank participants [35] and 4502 whole blood proteins from the deCODE genetics study [36] (n = 35,559; https://decode.com/summarydata; 1-Mar-2025).

### Interaction with thyroid function phenotypes

Free triiodothyronine (FT3) and thyroxine (FT4) did not qualify as mediators but caused a major departure of $b_2$ from its expected distribution (see Results). These traits may reflect thyroid gland issues. Given the relevance of hypothyroidism in the region, we analyzed these traits in detail. Because FT3 and FT4 levels were measured only when the thyroid stimulating hormone (TSH) level was < 0.4 or >3.8 μUI/mL, we started with a series of sensitivity analyses to exclude the presence of underlying sample stratification. First, we tested whether the distribution of the genotypes was different between individuals with and without measured FT3 and FT4, using a two-sided Wilcoxon test at a significance level of 0.05. Second, to verify whether there was any geographical cluster of individuals with extreme TSH levels (and so measured FT3 and FT4 levels), which might have implicated population stratification, we further adjusted the ln(eGFRcrea)-SNP association model in step 1 for municipality of residence.

After excluding the presence of sample stratification, we tested the interaction between the SNPs and TSH levels, which were measured in the whole sample. We excluded 416 individuals reporting any condition among thyroid cancer (n = 16), kidney cancer (n = 1), goiter (n = 277), having undergone surgery to the thyroid gland (n = 312), or having both missing TSH measurement and therapy information (n = 4), leaving 9730 individuals for the interaction analysis. Regression models included both the main and the interaction effect terms and were adjusted for the same covariates used in the step 1 model. In addition to quantitative TSH levels, we also tested interaction with broadly defined hyper- and hypothyroidism. Hyperthyroidism was defined as a TSH level of <0.4 μUI/mL or use of thiamazole (n = 3) or propylthiouracil (n = 1). Hypothyroidism was defined as TSH levels >3.8 μUI/mL or reported use of levothyroxine sodium (n = 512). No other thyroid-related treatment was reported. The statistical significance level for interaction testing was set at 0.0045 = 0.05/11 loci.

## Results

In the study sample, the median age was 46.5 years (interquartile range, IQR: 32.8, 57.6), females were 55%, and the median eGFRcrea level was 92.2 (IQR: 81.3, 103.2) ml/min/1.73m² (Table 1; additional clinical characteristics are reported in S1 Table in S2 File). The genomic inflation factor λ was 1.02 (S3 Fig in S1 File), supporting appropriate modeling of the sample structure. In the CHRIS study, 11 of the 147 CKDGen loci were associated with eGFRcrea (1-sided P-value between $3.34 \times 10^{-4}$ and $4.51 \times 10^{-7}$; Table 2; S2 Table in S2 File). For each such locus we retained all variants in LD ($r^2 > 0.8$) with the lead SNP, totaling 163 mostly intronic or non-coding exonic variants over the 11 loci (S3 Table in S2 File). All 11 loci exhibited very similar LD pattern in CHRIS as in CKDGen (S4 Fig in S1 File), consistent with shared European-ancestry background (Fig 2A). Among them, variants' effect magnitude at *CASZ1*, *DDX1*, *PIP5K1B*, *GAB2*, and *IGF1R* was significantly larger in CHRIS than in CKDGen (Fig 2B; Table 2). Consistently, they explained a larger proportion of ln(eGFRcrea) variance (Fig 2C). The CHRIS-to-CKDGen effect ratio varied between 1 and 5.4 and was not entirely explained by MAF differences (Fig 2D; S3 Table in S2 File). Three broad groupings characterized by MAF and effect size differences could be distinguished:

1. at *UMOD-PDILT*, *SLC34A1* and *SHROOM3*, we observed similar effect magnitude, regardless whether MAF was similar or lower in CHRIS, consistent with the fact that these loci were amongst the ones identified by the earliest GWAS thanks to their large effects, counterbalancing the still relatively limited sample sizes and the coarser genomic imputation [37,38];

**Table 1. Main characteristics of the 10,146 individuals included in this analysis. Additional clinical characteristics are described in S1 Table in S2 File.**

| Participants' characteristics | Median (IQR) or N (%) |
|---|---|
| Age, years | 46.5 (32.8–57.6) |
| Females | 5,585 (55.1%) |
| eGFRcrea, ml/min/1.73 m² (n = 10,141) | 92.2 (81.3–103.2) |
| Mg, mg/dl (n = 10,143) | 2.0 (1.9–2.1) |
| aPTT, seconds (n = 6,369) | 29.9 (28.6–31.4) |
| TSH, µUI/mL (n = 10,142) | 1.38 (0.98–1.93) |
| FT3, pg/ml (n = 466) | 3.00 (2.70–3.30) |
| FT4, ng/dl (n = 469) | 0.98 (0.87–1.08) |
| Hyperthyroidism (TSH < 0.4 or drug treatment) | 172 (1.7%) |
| Normal (0.4 < TSH < 3.8 and no drug treatment) | 8,972 (88.5%) |
| Hypothyroidism (TSH > 3.8 or drug treatment) | 998 (9.8%) |

Abbreviations: IQR, Interquartile range; TSH, Thyroid-stimulating hormone; FT3, Free triiodothyronine; FT4, Free thyroxine; Mg, serum magnesium; aPTT, activated partial thromboplastin time.

2. at *DAB2*, *TMEM60*, *PIP5K1B*, *GAB2*, and *IGF1R*, MAF was similar or slightly larger in CHRIS (effects in CHRIS were 1.3-to-5 times larger than in CKDGen);

3. at *STC1*, *DDX1*, and *CASZ1*, we observed lower MAF in CHRIS and about 1.8-to-5.4 times larger effects in CHRIS.

## Mediation analysis

To identify possible reasons for observed larger effects, we conducted mediation analysis of the association between eGFRcrea and each of the 163 variants at the 11 loci across 70 quantitative health traits available in CHRIS (S1 Table in S2 File). Adjusting the eGFRcrea-SNP association for each trait in turn (step 2 of the mediation analysis; see Methods) resulted in a substantial change of the eGFRcrea-SNP association coefficient in 4-to-12 variant-trait associations per locus (Fig 3). As expected, adjustment for the positive control SCr resulted in the nearly complete knockdown of the SNP effect at all loci. Often, the effect change happened when adjusting for traits closely related to kidney function such as urate (*GAB2*, *IGF1R*, *PIP5K1B*, *SHROOM3*, and *STC1*), urinary creatinine (*SHROOM3*), and urinary albumin (*IGF1R*). Serum electrolytes such as magnesium (*SHROOM3*), corrected calcium (*PIP5K1B*), and sodium (*DDX1*, *GAB2*, *RSBN1L*, *SLC34A1*, and *STC1*) were also widely involved. Of note was the widespread effect of FT3, FT4, and the activated partial thromboplastin time (aPTT) across all loci. When adjusting for FT3 and FT4, the SNP effect on ln(eGFRcrea) was often completely altered. This behavior across all loci may suggest the presence of either a measurement artifact or a general related condition affecting the underlying population.

In the 3rd mediation analysis step, we evaluated whether the traits identified as effect modifiers in step 2 were associated with the SNP of which they modified the association with eGFRcrea (Table 3; S4 Table in S2 File): SNP rs3812036 at *SLC34A1* was associated with aPTT levels ($P = 3.54 \times 10^{-6}$) and five SNPs at *SHROOM3* were associated with serum magnesium levels ($P$-values between $5.33 \times 10^{-5}$ and $6.12 \times 10^{-5}$). Three of our associations reproduce previously reported associations found using Phenoscanner: at *SLC34A1*, Tang *et al.* [39] reported association of rs3812036 with aPTT ($P = 2.88 \times 10^{-18}$), and at *SHROOM3*, Meyer *et al.* [40] reported associations of magnesium with rs4859682 ($P = 2.39 \times 10^{-9}$) and rs13146355 ($P = 6.27 \times 10^{-13}$).

All SNP-trait associations that successfully passed the 2nd mediation analysis step were also interrogated against the Phenoscanner, GWAS Catalog, and ThyroidOmics consortium results (Methods) to identify associations that might have been

Table 2. The 11 loci identified from the CKDGen Consortium GWAS that showed significant effects in CHRIS. Loci were selected if any variant in strong LD with the CKDGen lead variant was significantly associated with ln(eGFRcrea) in CHRIS. Consequently, for each locus, we report either the CKDGen lead variant (if this was significantly associated with ln(eGFRcrea) in CHRIS) or the CKDGen variant, in LD with the lead variant, which was most strongly associated with ln(eGFR-crea) in CHRIS (see Methods).

| Locus (gene name) | CKDGen lead SNP | Chr:Pos (build 38) | EA/OA | CKDGen Trans-Ancestry[1] | | | CKDGen EUR[1] | | | CHRIS | | | |
|---|---|---|---|---|---|---|---|---|---|---|---|---|---|
| | | | | EAF | b(SE) | P-value | EAF | b(SE) | P-value | EAF | b(SE) | 1-sided P-value | 2-sided P-value |
| CASZ1 | rs74748843 | 1:10,670,853 | T/C | 0.071 | −0.00480 (0.00078) | $7.84 \times 10^{-10}$ | 0.021 | −0.00598 (0.00123) | $1.09 \times 10^{-6}$ | 0.016 | −0.02776 (0.00806) | $2.85 \times 10^{-4}$ | $5.70 \times 10^{-4}$ |
| DDX1 | rs807624 | 2:15,642,347 | G/T | 0.580 | −0.00320 (0.00029) | $8.28 \times 10^{-28}$ | 0.660 | −0.00338 (0.00035) | $1.70 \times 10^{-21}$ | 0.701 | −0.00886 (0.00221) | $3.00 \times 10^{-5}$ | $5.99 \times 10^{-5}$ |
| SHROOM3 | rs28817415 | 4:76,480,299 | T/C | 0.400 | −0.00730 (0.00029) | $3.17 \times 10^{-137}$ | 0.440 | −0.00744 (0.00034) | $7.32 \times 10^{-109}$ | 0.428 | −0.00840 (0.00214) | $4.21 \times 10^{-5}$ | $8.42 \times 10^{-5}$ |
| DAB2 | rs10062079* | 5:39,393,631 | A/G | 0.039 | −0.00480 (0.00029) | $2.07 \times 10^{-60}$ | 0.430 | −0.00549 (0.00035) | $6.19 \times 10^{-56}$ | 0.453 | −0.00721 (0.00211) | $3.24 \times 10^{-4}$ | $6.49 \times 10^{-4}$ |
| SLC34A1 | rs3812036 | 5:177,386,403 | T/C | 0.260 | −0.00650 (0.00039) | $2.96 \times 10^{-62}$ | 0.260 | −0.00687 (0.00040) | $2.31 \times 10^{-67}$ | 0.240 | −0.00836 (0.00245) | $3.27 \times 10^{-4}$ | $6.55 \times 10^{-4}$ |
| TMEM60 | rs57514204* | 7:77,714,744 | T/C | 0.380 | −0.00340 (0.00029) | $3.53 \times 10^{-31}$ | 0.410 | −0.00332 (0.00034) | $1.48 \times 10^{-22}$ | 0.439 | −0.00723 (0.00212) | $3.34 \times 10^{-4}$ | $6.67 \times 10^{-4}$ |
| STC1 | rs819196* | 8:23,885,208 | T/A | 0.410 | −0.00410 (0.00029) | $1.47 \times 10^{-44}$ | 0.450 | −0.00406 (0.00034) | $1.43 \times 10^{-33}$ | 0.437 | −0.00718 (0.00210) | $3.21 \times 10^{-4}$ | $6.41 \times 10^{-4}$ |
| PIP5K1B | rs2039424 | 9:68,817,258 | G/A | 0.360 | −0.00440 (0.00029) | $4.75 \times 10^{-51}$ | 0.380 | −0.00483 (0.00035) | $1.12 \times 10^{-42}$ | 0.421 | −0.01042 (0.00212) | $4.51 \times 10^{-7}$ | $9.01 \times 10^{-7}$ |
| GAB2 | rs7113042* | 11:78,324,786 | A/G | 0.740 | −0.00270 (0.00039) | $4.62 \times 10^{-12}$ | 0.830 | −0.00271 (0.00045) | $1.86 \times 10^{-9}$ | 0.823 | −0.01096 (0.00286) | $6.41 \times 10^{-5}$ | $1.28 \times 10^{-4}$ |
| IGF1R | rs59646751 | 15:98,733,292 | T/G | 0.300 | −0.00230 (0.00029) | $3.97 \times 10^{-15}$ | 0.310 | −0.00203 (0.00036) | $2.47 \times 10^{-8}$ | 0.321 | −0.00909 (0.00223) | $2.32 \times 10^{-5}$ | $4.63 \times 10^{-5}$ |
| UMOD-PDILT | rs77924615 | 16:20,381,010 | G/A | 0.800 | −0.00980 (0.00039) | $4.38 \times 10^{-139}$ | 0.800 | −0.00958 (0.00044) | $1.52 \times 10^{-104}$ | 0.826 | −0.01030 (0.00281) | $1.27 \times 10^{-4}$ | $2.53 \times 10^{-4}$ |

[1] Trans-ancestry and European ancestry CKDGen results after subtracting the effects of 4661 CHRIS participants included in the CKDGen meta-analysis.

Abbreviations: Chr, chromosome; Pos, position; EA, effect allele; OA, other allele; EAF, effect allele frequency; b, coefficient of association; SE, standard error of b.

*Variant in strong LD with the CKDGen lead variant: in these cases, the variant associated with ln(eGFRcrea) in CHRIS was not the lead variant but one in strong LD with it. See S2 Table in S2 File for lead variant comparisons.

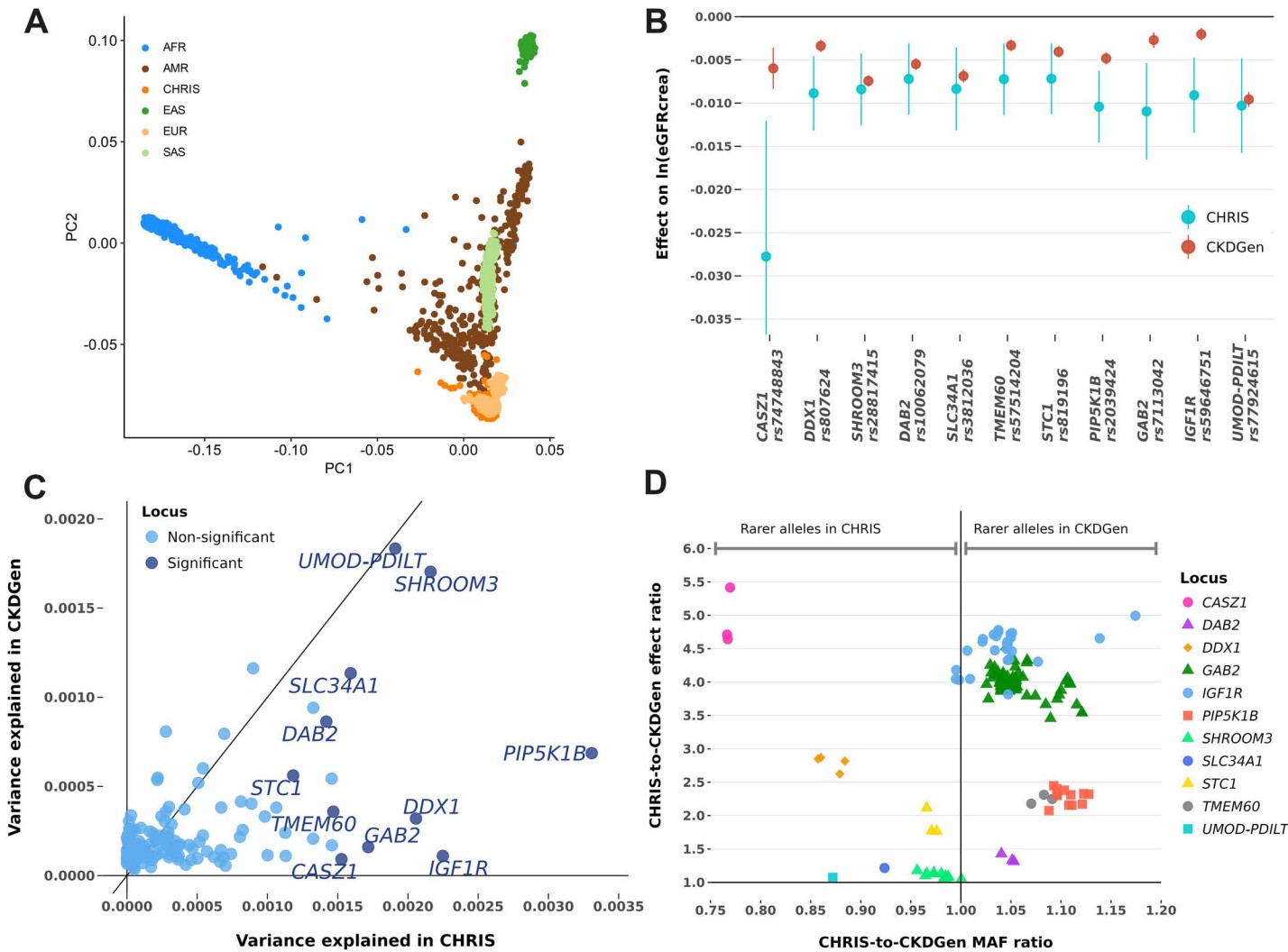

**Fig 2. CHRIS study results in the context of European-ancestry GWAS meta-analyses. Panel A:** Genetic overlap between CHRIS and 1000 Genomes project samples, showing that the CHRIS study matches the European ancestry (more extensive comparisons are provided in S6 Fig in S1 File). Continental origins are abbreviated as: AFR, Africa; AMR, Americas; EAS, East Asia; EUR, Europe; SAS, South Asia. **Panel B**: Effects of the lead variants at the 11 loci on eGFRcrea, in CKDGen-EUR and CHRIS, showing larger effects in CHRIS for variants at *CASZ1*, *DDX1*, *PIP5K1B*, *GAB2*, and *IGF1R*. Effects are meant per copy of the effect allele on the natural logarithm of eGFRcrea in ml/min/1.73m² (y-axis), following harmonization to the eGFRcrea-lowering allele. 95% confidence intervals are reported. For *CASZ1*, *DDX1*, *PIP5K1B*, *GAB2*, and *IGF1R* the magnitude of the effect in CHRIS was larger than in CKDGen. **Panel C**: Comparison of the ln(eGFRcrea) variance explained in CHRIS versus CKDGen-EUR by the lead variants at each of the 147 loci. The 11 loci significantly associated with ln(eGFRcrea) in CHRIS are colored in darker blue and labeled by gene name. **Panel D**: Ratio between CHRIS and CKDGen minor allele frequencies (MAF, x-axis) plotted against the ratio between CHRIS and CKDGen effect estimates (y-axis), for all 163 variants at the 11 loci that were significantly associated with eGFRcrea in CHRIS. Effects are expressed in terms of change of the natural logarithm of eGFRcrea in ml/min/1.73m² per copy of the effect allele.

missed by the 3rd step of the mediation analysis in CHRIS due to lack of power (S4 Table in S2 File). In addition to supporting our findings at *SHROOM3* and *SLC34A1*, this interrogation also revealed associations between variants at *IGF1R* and serum urate, unraveling how this additional mediation was likely missed by our investigation due to lack of power.

At *SHROOM3*, the partial mediation of Mg corresponded to about 11% effect attenuation (Table 3). eQTL investigation across all genes located in the *SHROOM3* locus did not identify any genome-wide significant association with *SHROOM3*

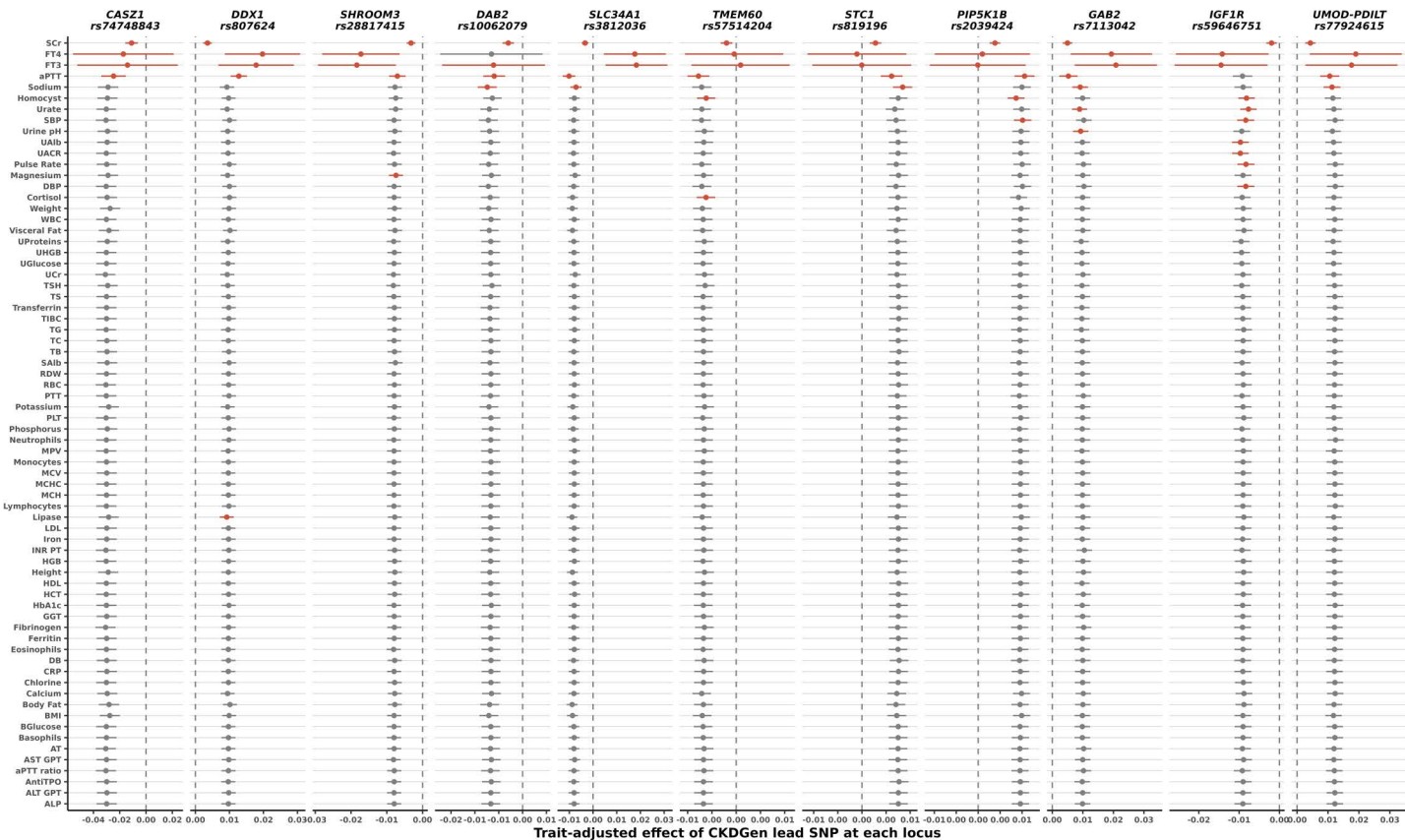

**Fig 3. Results of the 2nd step of the mediation analysis for the most associated variants at each locus.** SNP effect size and their 95% confidence intervals are reported. Effects are expressed in terms of change of the natural logarithm of eGFRcrea in ml/min/1.73m² per copy of the effect allele. Large departures from the expected effect size ('outlier', see Methods) are indicated in red. The vertical dashed lines identify the null. While adjustment for most of the 70 traits did not alter the locus-eGFRcrea association (grey dots), large effect changes were observed when adjusting for SCr (positive control; all loci), FT3 (all loci), FT4 (10 loci), aPTT (10 loci), sodium (5 loci), homocysteine (3 loci), SBP (2 loci), and urate (2 loci), as well as cortisol, DBP, lipase, magnesium, pulse rate, UACR, urinary albumin and urine pH (1 locus each). These traits were candidate to be classified as mediators at the respective loci, as tested in the 3rd step of the mediation analysis.

expression (Table 4), as it would be expected given the developmental nature of the gene [41,42]. However, *SHROOM3* resulted being the most associated gene in the kidney tissue. The variants involved in the mediation were also associated with cystatin C protein levels and with other kidney-function relevant proteins such as NAR3 and CXCL11 (I-TAC) [43] (Table 4).

At *SLC34A1*, the partial mediation of aPTT corresponded to a 21% larger effect of the SNP on ln(eGFRcrea). eQTL investigation highlighted association with *RGS14* expression across most tissues (Table 4). The involved genetic variant was an eQTL for coagulation factor 12 (F12) in the esophagous mucosa and a pQTL for kininostatin, a fragment of kininogen 1 (KNG1), and coagulation factor 2 (F2), all involved in the coagulation cascade (Table 4).

### Interaction with thyroid dysfunction

Despite the pervasive effect change caused by FT3 and FT4 in step-2 analysis, these two traits did not result in significant step-3 associations and so did not qualify as mediators. However, motivated by the fact that hypothyroidism is amongst the leading causes of hospitalization in the region [44], thyroid problems are common in Alpine areas, and the FT3 and

Table 3. Results of the mediation analysis at *SHROOM3* and *SLC34A1*. Extensive mediation analysis results for all loci and all traits are reported in S4 Table in S2 File.

| Locus (gene name) | rsID | Chr:Pos (build 38) | EA/OA | GWAS Step 1: association with ln(eGFRcrea) | | Trait | Step 2: association with ln(eGFRcrea) adjusted for the mediator | | % Change | Step 3: association with mediator in this study | | Step 4: association with mediator adjusted for ln(eGFRcrea) | |
|---|---|---|---|---|---|---|---|---|---|---|---|---|---|
| | | | | b(SE) | P | | b(SE) | P | | b(SE) | P | b(SE) | P |
| *SHROOM3* | rs28394165 | 4:76472865 | C/T | −0.00841 (0.00214) | $8.35 \times 10^{-5}$ | Mg | −0.00744 (0.00196) | $1.51 \times 10^{-4}$ | −12 | 0.00826 (0.00204) | $5.33 \times 10^{-5}$ | 0.00773 (0.00204) | $1.54 \times 10^{-4}$ |
| *SHROOM3* | rs10025351 | 4:76472942 | T/C | −0.00840 (0.00214) | $8.51 \times 10^{-5}$ | Mg | −0.00743 (0.00196) | $1.53 \times 10^{-4}$ | −12 | 0.00823 (0.00204) | $5.73 \times 10^{-5}$ | 0.00769 (0.00204) | $1.65 \times 10^{-4}$ |
| *SHROOM3* | rs28817415 | 4:76480299 | T/C | −0.00840 (0.00214) | $8.42 \times 10^{-5}$ | Mg | −0.00744 (0.00196) | $1.51 \times 10^{-4}$ | −12 | 0.00823 (0.00204) | $5.68 \times 10^{-5}$ | 0.00770 (0.00204) | $1.64 \times 10^{-4}$ |
| *SHROOM3* | rs4859682 | 4:76489165 | A/C | −0.00840 (0.00210) | $6.59 \times 10^{-5}$ | Mg | −0.00750 (0.00193) | $1.02 \times 10^{-4}$ | −11 | 0.00812 (0.00201) | $5.49 \times 10^{-5}$ | 0.00758 (0.00201) | $1.63 \times 10^{-4}$ |
| *SHROOM3* | rs13146355 | 4:76490987 | A/G | −0.00840 (0.00214) | $8.38 \times 10^{-5}$ | Mg | −0.00749 (0.00196) | $1.33 \times 10^{-4}$ | −11 | 0.00819 (0.00204) | $6.12 \times 10^{-5}$ | 0.00765 (0.00204) | $1.77 \times 10^{-4}$ |
| *SLC34A1* | rs3812036 | 5:177386403 | T/C | −0.00836 (0.00245) | $6.55 \times 10^{-4}$ | aPTT | −0.01013 (0.00269) | $1.64 \times 10^{-4}$ | 21 | −0.23110 (0.04980) | $3.54 \times 10^{-6}$ | −0.23046 (0.04986) | $3.88 \times 10^{-6}$ |

Abbreviations: Chr, chromosome; Pos, position; EA, effect allele; OA, other allele; b, coefficient of association; SE, standard error of b; Mg, Magnesium; aPTT, activated Partial Prothrombin Time.

Table 4. eQTL and pQTL interrogation of *SHROOM3* and *SLC34A1* variants implicated by mediation analysis. All reported genes and proteins were associated with the respective variants at genome-wide significance level, except for the Susztak lab eQTLs [34], for which we report all results with P-value<5×10⁻⁵. Extensive results are reported in S8-S11 Tables in S2 File.

| Locus | rsID Chr:Pos* | EA | eGFR-crea | Mediator | GTEx v10 non-kidney eQTLs gene name (no. of tissues)** | Human Kidney eQTL data [34] gene name (P-value) | UK Biobank pQTLs [35] gene (protein) | deCODE pQTLs [36] gene (protein) |
|---|---|---|---|---|---|---|---|---|
| *SHROOM3* | rs28394165 4:76472865 | C | ↓ | ↑Mg | ↓ FAM47E (18)<br>↓ CCDC158 (7)<br>↑ ENSG00000289515*** (6)<br>↑ STBD1 (4) | ↑SHROOM3 ($P$=6.4×10⁻⁷)<br>↑STBD1 ($P$=7.2×10⁻⁶) | ↑NPC2 (NPC2) | |
| *SHROOM3* | rs10025351 4:76472942 | T | ↓ | ↑Mg | ↓ FAM47E (18)<br>↓ CCDC158 (7)<br>↑ ENSG00000289515 (6)<br>↑ STBD1 (4) | ↑SHROOM3 ($P$=7.9×10⁻⁷)<br>↑STBD1 ($P$=8.5×10⁻⁶) | | ↑ CST3 (Cystatin C)<br>↑ CXCL11 (I-TAC)<br>↑ NAAA (ASAHL)<br>↑ ART3 (NAR3) |
| *SHROOM3* | rs28817415 4:76480299 | T | ↓ | ↑Mg | ↓ FAM47E (18)<br>↓ CCDC158 (7)<br>↑ ENSG00000289515 (6)<br>↑ STBD1 (4) | | ↑ COL18A1 (Endostatin)<br>↑ EFNA1 (Ephrin-A1)<br>↑ MIA (MIA)<br>↑ RNASE4 (RNAS4)<br>↑ SPINK1 (TATI) | ↑ CST3 (Cystatin C)<br>↑ CXCL11 (I-TAC)<br>↑ NAAA (ASAHL)<br>↑ ART3 (NAR3) |
| *SHROOM3* | rs4859682 4:76489165 | A | ↓ | ↑Mg | ↓ FAM47E (19)<br>↓ CCDC158 (8)<br>↑ ENSG00000289515 (6)<br>↑ STBD1 (4) | ↑SHROOM3 ($P$=4.2×10⁻⁷)<br>↑STBD1 ($P$=8.2×10⁻⁶) | ↑ CCN5 (WISP-2)<br>↑ IGFBP4 (IGFBP-4)<br>↑ NPPC (Natriuretic Peptide C-Type)<br>↑ RNASE1 (RNase 1) | ↓ PRCP (Prolylcarboxy-peptidase)<br>↑ CST3 (Cystatin C)<br>↑ CXCL11 (I-TAC)<br>↑ NAAA (ASAHL)<br>↑ ART3 (NAR3) |
| *SHROOM3* | rs13146355 4:76490987 | A | ↓ | ↑Mg | ↓ FAM47E (18)<br>↓ CCDC158 (7)<br>↑ ENSG00000289515 (6)<br>↑ STBD1 (4) | ↑SHROOM3 ($P$=5.4×10⁻⁷)<br>↑STBD1 ($P$=1.2×10⁻⁵) | ↑ CST3 (Cystatin C)<br>↑ HSPG2 (Perlecan)<br>↑ IGFBP6 (IGFBP-6)<br>↑ NPDC1 (NPDC1)<br>↑ OGN (MIME)<br>↑ RELT (RELT)<br>↑ SHISA5 (SHISA5)<br>↑ TNFRSF19 (TAJ) | ↓ PRCP (Prolylcarboxy-peptidase)<br>↑ CST3 (Cystatin C)<br>↑ CXCL11 (I-TAC)<br>↑ NAAA (ASAHL)<br>↑ ART3 (NAR3) |
| *SLC34A1* | rs3812036 5:177386403 | T | ↓ | ↓ aPTT | ↓ RGS14 (33)<br>↓ MXD3 (Tibial nerve; Muscoskeletal)<br>↓ FGFR4 (Tibial nerve)<br>↓ F12 (Esophagus Mucosa) | | | ↓ PLXDC1 (PXDC1)<br>↓ PTH (PTH)<br>↓ DPEP2 (DPEP2)<br>↑ LMAN2 (Lectin, mannose binding 2)<br>↑ KNG1 (Kininostatin)<br>↑ CTSV (Cathepsin V)<br>↑ F2 (Thrombin)<br>↑ SOD2 (Mn SOD) |

**Abbreviations:** Chr, chromosome; Pos, position; EA, effect allele; Mg, Magnesium; aPTT, activated Partial Prothrombin Time.

**Symbols:** ↓ or ↑, effect allele associated with lower/ higher levels of the marker, respectively.

*Build 38; **Actual tissue reported when number of tissues was ≤2; ***long-non-coding RNA Lnc-CCDC158-4.

FT4 adjustment affected the SNP-eGFRcrea association at all 11 loci, we considered it reasonable to evaluate a potential role of thyroid dysfunction as a modifier of the SNP-eGFRcrea relation.

Preliminarily, given that in the CHRIS study FT3 and FT4 were measured only in individuals with TSH levels above or below specific thresholds, we conducted sensitivity analyses to exclude the presence of artifacts. We did not observe any genotype stratification by measured versus unmeasured FT3 and FT4 levels (S5 Table in S2 File). Adjustment for municipality of residence, to exclude the presence of local clusters of thyroid dysfunction, did not alter the results either (S5 Fig in S1 File).

We then tested for the presence of linear interaction with TSH levels, which identified a significant interaction at *STC1* SNPs rs819185 (P = 0.00177) and rs819196 (P = 0.00154) both below the multiple testing threshold of 0.05/11 loci (Fig 4; S6 Table in S2 File). Since for most loci the adjustment both for FT3 and for FT4 were causing an increase of the effect magnitude, we also postulated a U-shaped interaction model, with the SNP effects on ln(eGFRcrea) being larger both in hyper- and hypothyroidism. We classified individuals as healthy (88.5%), with hypothyroidism (9.8%) or with hyperthyroidism (1.7%; Table 1). Results did not support the presence of interaction with these two conditions at any locus after multiple testing control, despite nominally significant P-values observed for SNP interaction with hyperthyroidism at *SHROOM3* (P = 0.03) and with hypothyroidism at *PIP5K1B* (P = 0.03) and *GAB2* (P = 0.04; S7 Table in S2 File).

## Discussion

Out of 147 loci known for their association with kidney function, we identified 11 loci that were more strongly associated with eGFRcrea in the CHRIS study sample than in average European-ancestry population samples. Observed differences

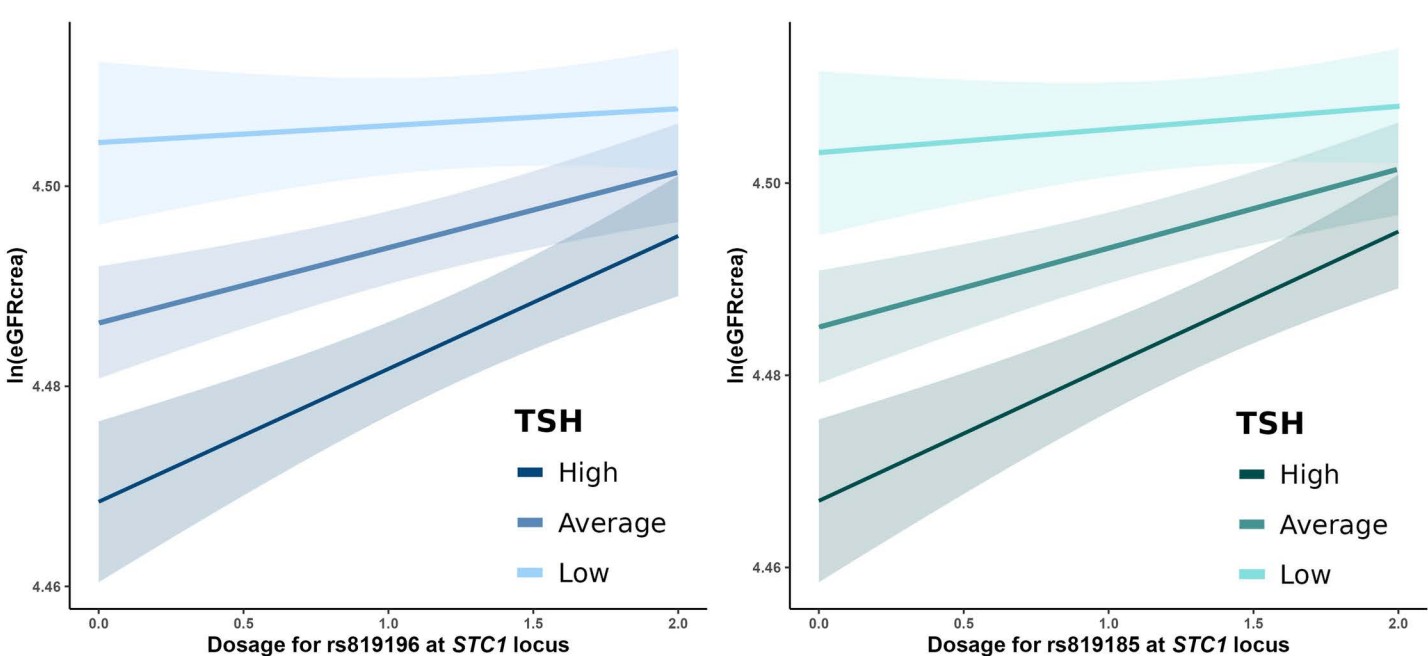

**Fig 4. Interaction of ln(eGFRcrea)-associated variants at *STC1* with TSH levels.** The effect of *STC1* SNPs rs819196 (P = 0.00154; left panel) and rs819185 (P = 0.00177; right panel) on eGFRcrea was stronger at lower TSH levels. *P*-values for interaction were 0.00177 and 0.01154 for rs819185 and rs819196, respectively. Using *lm* function, ln(eGFRcrea) was regressed on SNP dosage, TSH levels, their interaction, and covariates (sex, age, and the first 10 genetic principal components). The plots, generated using the *interact_plot* function in the R 'interactions' package (https://doi.org/10.32614/CRAN.package.interactions), show predicted ln(eGFRcrea) values with 95% confidence intervals for TSH levels one standard deviation below (low), above (high), and near the mean (average).

in effect magnitude could not be explained by MAF differences. Extensive mediation analysis identified serum magnesium and aPTT as partial mediators of the association of eGFRcrea with variants at *SHROOM3* and *SLC34A1*, respectively. SNP-by-TSH interaction analysis highlighted that the effect of variants at *STC1* on eGFRcrea may vary by TSH levels.

*UMOD-PDILT*, *SHROOM3*, and *SLC34A1* explained about the same fraction of eGFRcrea variance in CHRIS and CKDGen. This could have been expected as these historical loci were already detected by old GWAS studies with relatively small sample sizes [37,38]. For all other loci, the significant association was due to an interplay between allele frequency differences and effect estimate differences. Fig 2 demonstrates that the effect differences were probably more relevant than the allele frequency differences. But the question remains as to why precisely these and no other loci showed such large effects in CHRIS. SNPs may show larger effects at lower MAF. This was observed at *DDX1* and *CASZ1*, whose larger effects corresponded to smaller CHRIS-to-CKDGen MAF ratio. However, the MAF difference was unrelated to the effect magnitude at *GAB2*, *TMEM60*, *PIP5K1B*, and *IGF1R*: in these cases, MAF was similar or even smaller in CKDGen than in CHRIS (Fig 2D). The presence of un-modeled sample structure causing genomic inflation can be excluded as relatedness was appropriately modeled. We can also exclude the presence of genetic admixture as the CHRIS study was conducted in a small area, on a local population of homogeneous ancestry [15,45–47]. LD structure differences between the local and the overall population can also be excluded as regional association plots showed similar shapes between CHRIS and CKDGen. Epistatic gene-gene interaction would be a further possibility that cannot be easily verified. The most plausible remaining reasons would be the presence of local effect modifiers or gene-by-environment interaction. Mediation analysis based on commonly measured biochemical and anthropometric traits provided limited answers in this respect.

Statistical mediation observed at *SHROOM3* and *SLC34A1* is compatible with different causal scenarios (Fig 5). First, given several established connections between both loci and eGFRcrea, it looks implausible that genes and eGFRcrea

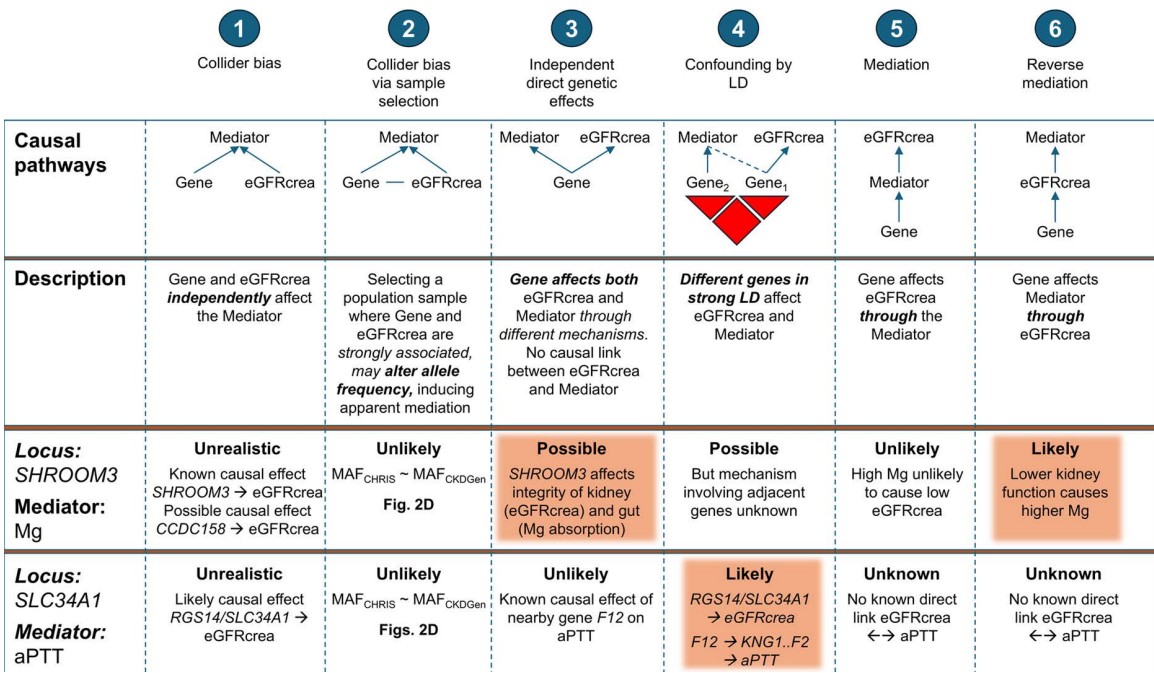

**Fig 5. Causal scenarios underlying the observed statistical mediations.** The observed mediations of Mg on the *SHROOM3*-eGFRcrea association and aPTT on the *SLC34A1*-eGFRcrea association, may be the result of different causal pathways, including the presence of collider bias, confounding, or genuine biological mediation. The six scenarios illustrated here are not mutually exclusive and probably do not cover the spectrum of possibilities.

may independently affect the mediators, generating apparent mediation (Fig 5, scenario 1). Moreover, given similar allele frequencies between CHRIS and CKDGen, collider bias acting through population selection seems implausible (Fig 2A and 2D). At *SHROOM3*, apparent mediation due to LD confounding is possible: there are several haplotypes spanning all genes in the locus that were jointly associated with eGFRcrea and Mg [43]. However, it is unclear which of such genes might affect Mg without jointly affecting eGFRcrea. Known biology should also bring to exclude a causal effect of Mg on eGFRcrea. For *SHROOM3*, the most likely scenario looks like that of *SHROOM3* having independent causal effects on eGFRcrea and Mg levels. Genetic variants at *SHROOM3* have been associated with CKD [8,9], reduced eGFRcrea [9], increased albumin-to-creatinine ratio [10], and low serum magnesium levels [11]. *SHROOM3* is necessary to maintain the glomerular filtration barrier integrity [41]. Variants in this gene have been shown to be associated with increased risk of CKD through disruption of the transcription factor TCF7L2 in podocyte cells [48]. The nearby *CCDC158* is involved in renal proximal tubular endocytosis and is expressed in renal proximal tubular cells and in glomeruli of individuals free from nephrocalcinosis [49]. There is thus support for a causal role of the *SHROOM3* locus on kidney function, at least through *SHROOM3* itself, but not excluding *CCDC158*. On the magnesium side, one can note that *SHROOM3* is also expressed in the epithelia morphogenesis of the gut [50], which has a crucial role in Mg absorption and regulation. This evidence would be compatible with a previous observation that Mg association with the *SHROOM3* locus did not change when adjusting for eGFRcrea, suggesting pleiotropic independent effects [40]. However, being serum magnesium a poor marker of dietary magnesium [51] this hypothesis should be considered with caution. On the other hand, our observation of a partial mediation (Mg adjustment caused 11% reduction of *SHROOM3* effect on eGFRcrea), would be compatible with the kidney's role on Mg regulation, through excretion and reabsorption. By this reasoning, high Mg levels could be partially determined by long-term lower kidney function levels due to *SHROOM3* variations affecting embryonal kidney development. Altogether, combining Fig 5's possible scenario 3 and likely scenario 6 would imply the partial mediation as the result from both a direct effect of *SHROOM3* on Mg and an indirect effect of *SHROOM3* on Mg through lower kidney function.

At *SLC34A1*, confounding by LD seems the most likely possibility. Both *RGS14* [52] and *SLC34A1* [53] are involved in phosphate handling that is critically dealt by the kidney, also in response to parathyroid hormone (PTH) regulation, as confirmed by our pQTL lookup that identified association with PTH in the deCODE proteomics data. The association with eGFRcrea [38] likely reflects causal involvement of this locus in kidney function regulation. In addition, the locus was previously associated with aPTT in 9240 European ancestry individuals [39] including participants from the same area where the CHRIS study was sampled. By quantifying the clotting time from the activation of factor XII (*F12*) until the formation of a fibrin clot, aPTT is a measure of the integrity of the intrinsic and common coagulation pathways [54]. The observed mediation, in which adjustment for aPTT increased the SNP-eGFRcrea association by 21%, is likely related to the proximity between *SLC34A1* and the coagulation factor gene *F12*. F12 initiates the coagulation cascade, which in turn involves kininogen 1 (KNG1) and coagulation factor 2 (F2), both of which were identified in our pQTL lookup. Results are thus compatible with the presence of common haplotypes partially tagging both *RGS14*/*SLC34A1* and *F12*, inducing partially overlapping effects on eGFRcrea and aPTT.

An additional, relevant finding was the presence of an interaction between two SNPs at *STC1* and TSH levels. This result deserves replication in independent populations both to validate it and to assess the degree of generalizability of our results, which might be conditioned on specific environmental exposures. *STC1* is chiefly expressed in the thyroid gland (https://gtexportal.org/home/gene/STC1) but also highly expressed in the kidney's collecting duct [55]. Evidence from a nomad population sample in Kenya, suggested that *STC1* is associated with urate and urea levels and is involved in the physiological response to dehydration and protein-rich diet [56]. The *STC1*-encoded protein, stanniocalcin-1, is involved in phosphate reabsorption in the kidney proximal tubules and in multiple pathophysiological mechanisms including ischemic kidney injury [57]. Stanniocalcin-1 behaves like an endocrine hormone, with both autocrine and paracrine functions. Our finding of a SNP effect on eGFRcrea dependent on TSH levels would suggest that altered thyroid function levels might

affect Stanniocalcin-1 with consequent effect on kidney function phenotypes. While intriguing, this hypothesis should be verified by appropriate studies.

Our analysis has strengths and limitations. The analysis of a local homogeneous population sample was at the same time the most relevant strength and limitation of our investigation. While homogeneity may enhance the identification of specific genetic associations, it may also hinder the possibility to explain the underlying reasons of the observed larger effects compared to the average European ancestry population. The availability of a very large number of clinical parameters has allowed us to conduct an exhaustive and unbiased screening on potential mediation mechanisms. However, the analyzed traits lacked molecular specificity to identify underlying mechanisms. In addition to kidney function, eGFRcrea may partially reflect creatinine metabolism. Unfortunately, the CHRIS study did not measure alternative kidney function markers such as cystatin C or BUN. For this reason, we limited our analyses to loci whose kidney function relevance was already demonstrated by previous studies [17]. Assessing environmental exposures may also help explain the observed larger effects and would be ground for further research. Finally, our choice of adjustment for multiple testing is worth a brief reflection. We considered 147 loci that were already validated by at least one previous study through discovery and independent replication [17,38,58,59]. As such, they wouldn't need further validation and a nominal significance level would be commonly accepted for hypothesis testing. However, the aim of our analysis was to identify loci with substantial effects in the CHRIS study to allow conducting meaningful mediation analyses. For this reason, we choose to penalize the significance level by the number of independent loci. This seems a sensible choice also by the fact that 147 independent tests covered 98% of the total variability of the 6337 SNPs in strong LD observed overall.

In conclusion, we identified loci with particularly large effect on kidney function in this specific Alpine population sample, namely *CASZ1*, *IGF1R*, *GAB2*, and *DDX1*. Allele frequency and LD analyses did not explain the observed larger effects nor did the extensive mediation analyses conducted. On the other hand, we showed that the effects of *SHROOM3* and *SLC34A1* on eGFRcrea are partially mediated by serum magnesium and aPTT, respectively. Finally, the observed *STC1*-TSH interaction implicates thyroid function involvement, which might have been favored by the high burden of thyroid-related diseases in the region. Further investigations are warranted to link these findings to environmental and molecular characteristics of specific population samples that may elucidate mechanisms not otherwise identifiable in large genetic meta-analyses.

## Supporting information

**S1 File. Supplementary methods and figures.**
(DOCX)

**S2 File. Supplementary tables.**
(XLSX)

## Acknowledgments

We thank all study participants, the Healthcare System of the Autonomous Province of Bolzano-South Tyrol, and all Eurac Research staff involved in the study (https://www.eurac.edu/en/institutes-centers/institute-for-biomedicine/pages/acknowledgements). Bioresource Impact Factor Code: BRIF6107. We thank the Department of Innovation, Research, University and Museums of the Autonomous Province of Bozen/Bolzano for covering the Open Access publication costs.

## Author contributions

**Conceptualization:** Dariush Ghasemi-Semeskandeh, Cristian Pattaro.

**Data curation:** David Emmert, Eva König, Martin Gögele, Laura Barin, Christian Fuchsberger.

**Formal analysis:** Dariush Ghasemi-Semeskandeh.

**Funding acquisition:** Cristian Pattaro.

**Investigation:** Dariush Ghasemi-Semeskandeh, David Emmert.

**Project administration:** Peter P. Pramstaller, Cristian Pattaro.

**Supervision:** Cristian Pattaro.

**Visualization:** Dariush Ghasemi-Semeskandeh, Mohamed S. Sarhan.

**Writing – original draft:** Dariush Ghasemi-Semeskandeh, Cristian Pattaro.

**Writing – review & editing:** Dariush Ghasemi-Semeskandeh, David Emmert, Eva König, Luisa Foco, Martin Gögele, Mohamed S. Sarhan, Laura Barin, Ryosuke Fujii, Christian Fuchsberger, Dorien J.M. Peters, Cristian Pattaro.

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
