## [Decision Letter · Decision Letter 0]

Dear Dr. Ghasemi Semeskandeh,

Thank you for submitting your manuscript to PLOS ONE. After careful consideration, we feel that it has merit but does not fully meet PLOS ONE’s publication criteria as it currently stands. Therefore, we invite you to submit a revised version of the manuscript that addresses the points raised during the review process.

We look forward to receiving your revised manuscript.

Kind regards,

Andrew J. Mallett

Academic Editor

PLOS ONE

Journal Requirements:

“The CHRIS study was funded by the Department of Innovation, Research and University of the Autonomous Province of Bolzano-South Tyrol. This work was carried out within the TrainCKDis project, funded by the European Union’s Horizon 2020 research and innovation programme under the Marie Skłodowska-Curie grant agreement H2020-MSCA-ITN-2019 ID:860977 (TrainCKDis).”

“The CHRIS study was funded by the Department of Innovation, Research and University of the Autonomous Province of Bolzano-South Tyrol. This work was carried out within the TrainCKDis project, funded by the European Union’s Horizon 2020 research and innovation programme under the Marie Skłodowska-Curie grant agreement H2020-MSCA-ITN-2019 ID:860977 (TrainCKDis).”

4. We notice that your supplementary [figures/tables] are included in the manuscript file. Please remove them and upload them with the file type 'Supporting Information'. Please ensure that each Supporting Information file has a legend listed in the manuscript after the references list.

Reviewers' comments:

Reviewer's Responses to Questions

**Comments to the Author**

1. Is the manuscript technically sound, and do the data support the conclusions?

Reviewer #1: Yes

Reviewer #2: Yes

Reviewer #3: Yes

2. Has the statistical analysis been performed appropriately and rigorously?

Reviewer #1: Yes

Reviewer #2: Yes

Reviewer #3: Yes

3. Have the authors made all data underlying the findings in their manuscript fully available?

Reviewer #1: Yes

Reviewer #2: Yes

Reviewer #3: No

4. Is the manuscript presented in an intelligible fashion and written in standard English?

Reviewer #1: Yes

Reviewer #2: Yes

Reviewer #3: Yes

Reviewer #1: This is an interesting article which takes GWAS meta analysis data and undertakes a systematic mediation to look for quantitative traits that may underlie the SNPs leading to chronic kidney disease. Some additions are needed to maximise the utility of this analysis.

Page 4 - the rather convoluted text regarding replication needs to be revised / simplified. The authors say it is not a replication analysis but then refer to it as a replication analysis. I understand their meaning but it needs to be phrased more clearly.

The association of variants in SLC34A1 and aPTT - this needs to be dissected further, as the authors point out the snp is near to F12 which is more likely to be biologically relevant. Some eQTL or candidate cis-regulatory elements (cCREs) analysis for the SNPs of interest should be investigated (see Loeb GB et al Nature Genetics).

The PDILT snp is usually associated with UMOD - can this be mentioned clarified

The SHROOM3 findings related to Magnesium are interesting and perhaps a mechanistic diagram would help here to understand how SHROOM3 might be mediating magnesium regulation. Again for the SHROOM3 snp some QTL or candidate cis-regulatory elements (cCREs) analysis for the SNPs of interest should be investigated.

It would be useful to know if any of the SNPs are in coding regions.

The associations with abnormal thyroid function as modifiers are really interesting and that adds to the novelty of this study

Reviewer #2: Ghasemi-Semeskandeh et al., in this study of the CHRIS cohort (n=10,146), replicated associations for 11 kidney function loci, showing effect sizes up to 5.4 times larger than those observed in the CKDGen kidney function meta-GWAS involving 1 million individuals. Mediation analysis highlighted serum magnesium and activated partial thromboplastin time as partial mediators at SHROOM3 and SLC34A1, respectively. Additionally, thyroid function was identified as a genetic effect modifier, with significant SNP-by-TSH interactions at STC1. These findings underscore the value of individual population studies in elucidating genetic mechanisms underlying kidney function. However, I have several suggestions for the authors to consider:

1.Ethical Statement Placement: In the Methods section, I suggest beginning with the Ethical Statement, which is currently placed at the end of the section, for improved organization and clarity.

2.Selection of Loci: The authors selected 147 kidney function loci from over 264 independent loci reported in the CKDGen study. Please clarify how this number was reduced to 147 after excluding CHRIS samples from the GWAS summary statistics. Additionally, it is unclear how MetaSubtract was used to exclude CHRIS data from the summary statistics. Why not conduct the analysis solely on independent CHRIS subjects (n=~10,000), even if the sample size is smaller, to ensure cleaner results?

3.Correction for Multiple Testing: Did the authors account for multiple testing in the mediation analysis, given that 70 quantitative traits were assessed? While multiple testing correction for SNPs was mentioned, clarifying this point for the quantitative trait analysis would strengthen the study.

4.Evaluation of Collider Bias: Did the authors assess the potential for collider bias during mediation analysis, particularly for the quantitative traits? Addressing this would help validate the findings further.

5.Discrepancy in Variant Count: The authors mention testing 147 variants, but in the Results section (Page 9, Line 256), they state that 163 variants were tested at 11 loci. Are these independent SNPs? Please clarify this discrepancy and provide a detailed explanation.

6.Figure Legends: Some figures lack clear explanations in their legends, making it difficult to understand what tests were conducted. Providing detailed figure legends will significantly improve comprehension.

7.PCA Plot for Ancestry: To enhance clarity, consider including a PCA plot of the CHRIS cohort alongside the 1000 Genomes dataset to better illustrate ancestry composition.

Overall, addressing these points will enhance the clarity, organization, and rigor of the study.

Reviewer #3: The manuscript titled "Systematic mediation and interaction analyses of kidney function genetic loci in a general population study" explores the genetic factors influencing kidney function, specifically focusing on chronic kidney disease. The authors conducted a study using data from the Cooperative Health Research In South Tyrol (CHRIS) involving 10,146 participants to assess the association of 147 kidney function-related genetic loci with estimated glomerular filtration rate (eGFR) based on serum creatinine levels. The manuscript addresses a topic of great clinical importance, the relationship between genetic factors and kidney function, especially in a specific population context. The identification of genetic loci associated with glomerular filtration rate (GFR) may contribute to understanding the genetic basis of chronic kidney disease (CKD) and its implications for treatment and prevention.

Introduction

The introduction could be more robust in contextualizing the problem. It would be useful to include a literature review explaining the relevance of the genetic loci chosen and how they relate to kidney function and chronic kidney disease.

Statistical Issues:

The statistical analysis is adequate, but the choice to penalize the significance level for multiple tests must be clearly justified.

The use of a large and homogeneous sample, such as that of the CHRIS study, is a strength, as it allows for a more robust analysis of genetic associations. However, it is important to discuss the limitations of the methodology, such as the dependence on specific markers (such as creatinine) to estimate GFR and the need for validation in different populations.

Results:

It would be beneficial to include a section discussing the validation of the results in independent cohorts or the comparison with previous studies. This will help strengthen the credibility of the findings.

Discussion:

The discussion about the limitations of the study should be more comprehensive. Consider discussing the homogeneity of the sample and how this may affect the generalizability of the results.

Conclusion:

The conclusions are consistent with the results presented, but should emphasize the need for future investigations to explore the underlying mechanisms and applicability of the findings in clinical contexts.

**Do you want your identity to be public for this peer review?** For information about this choice, including consent withdrawal, please see our Privacy Policy

Reviewer #1: No

Reviewer #2: No

Reviewer #3: No

---

## [Author Response · Author response to Decision Letter 1]

13 Mar 2025

Point-by-point response to Journal Requirements

Journal Requirements:

Response: We revised the style of our manuscript, including file naming, ensuring that everything is now adherent to PLOS ONE’s requirements.

Response: Individual-level data used for this analysis are subject to restrictions due to the Italian personal data processing legislation that does not allow sharing of personal data without a data transfer agreement between the data controller and the single recipient (GDPR art. 26 and 28). For non-personal information, the ethical framework foresees that any use of the data must be in line with the participants’ consent. The study framework (approved by the local ethics committee) requires that the CHRIS Access Committee must review any sharing. Current analysis data can be requested with an application to the CHRIS Access Committee at access.request.biomedicine@eurac.edu.

Response: The data availability statement was already adherent to the limitations described above. Please, let us know if we should include the whole explanation why data are subject to restrictions.

“The CHRIS study was funded by the Department of Innovation, Research and University of the Autonomous Province of Bolzano-South Tyrol. This work was carried out within the TrainCKDis project, funded by the European Union’s Horizon 2020 research and innovation programme under the Marie Skłodowska-Curie grant agreement H2020-MSCA-ITN-2019 ID:860977 (TrainCKDis).”

Response: We updated the Acknowledgment section, which does not include any reference to funding anymore.

“The CHRIS study was funded by the Department of Innovation, Research and University of the Autonomous Province of Bolzano-South Tyrol. This work was carried out within the TrainCKDis project, funded by the European Union’s Horizon 2020 research and innovation programme under the Marie Skłodowska-Curie grant agreement H2020-MSCA-ITN-2019 ID:860977 (TrainCKDis).”

Response: We removed the funding statement from the manuscript. In the meantime, the CHRIS study has updated their funding and acknowledgement statements, requiring us to use the following funding statement:

The CHRIS Study was funded by the Autonomous Province of Bolzano-South Tyrol Department of Innovation, Research, University and Museums and supported by the European Regional Development Fund (FESR1157). This work was carried out within the TrainCKDis project, funded by the European Union’s Horizon 2020 research and innovation programme under the Marie Skłodowska-Curie grant agreement H2020-MSCA-ITN-2019 ID:860977 (TrainCKDis).

Response: Thank you! We have included all amended statements in this letter.

4. We notice that your supplementary [figures/tables] are included in the manuscript file. Please remove them and upload them with the file type 'Supporting Information'. Please ensure that each Supporting Information file has a legend listed in the manuscript after the references list.

Response: Every Supporting Information file is now listed, with a legend, in the manuscript, after the references list.

Point-by-point response to Reviewers

Reviewer #1:

This is an interesting article which takes GWAS meta analysis data and undertakes a systematic mediation to look for quantitative traits that may underlie the SNPs leading to chronic kidney disease. Some additions are needed to maximise the utility of this analysis.

Response: we thank the Reviewer for their positive assessment of our work. In the following, we respond to each specific issue.

1. Page 4 - the rather convoluted text regarding replication needs to be revised / simplified. The authors say it is not a replication analysis but then refer to it as a replication analysis. I understand their meaning but it needs to be phrased more clearly.

Response: We agree with the Reviewer that the use of the “replication” terminology was misleading. We re-wrote the rationale of the study (see Introduction, pages 3-4, page 85-91) which now reads as follows:

“We focused our investigation on 147 loci identified by a large GWAMA of the CKDGen Consortium with proven kidney function relevance [17]. In the cited work, the CKDGen identified 308 loci associated with eGFRcrea at genome-wide significance level; of them, 264 were replicated in an independent population study and thus validated; but given eGFRcrea might also reflect creatinine metabolism in addition to kidney function, the authors further filtered their results for direction-consistent association with blood urea nitrogen (BUN), an alternative marker of kidney function [17].”

and removed the concept of replication throughout.

2. The association of variants in SLC34A1 and aPTT - this needs to be dissected further, as the authors point out the snp is near to F12 which is more likely to be biologically relevant. Some eQTL or candidate cis-regulatory elements (cCREs) analysis for the SNPs of interest should be investigated (see Loeb GB et al Nature Genetics).

Response: We thank the Reviewer for their observation. We have now generated a new Table 4 including expression quantitative trait loci (eQTLs) from the GTEx Consortium version 10 and kidney tissue from the Susztak laboratory dataset, and protein QTLs (pQTLs) from the UK Biobank and DECODE genetics databases.

Results illuminate further the reasons of the observed mediation. From the kidney function perspective, the most associated gene was RGS14, which plays a role in phosphate metabolism by the kidney. Thus, there are two genes under the same signal both involved in kidney phosphate handling: RGS14 and SLC34A1. On the coaugulation side, we have identified an eQTL for F12 in the esophagous mucosa plus pQTLs for kininogen 1 (KNG1) and coagulation factor 2 (F2).

We thus speculate that the observed statistical mediation might be the result of confounding by LD, with nearby genes acting independently on parallel pathways: one with causal effects on kidney function and one for the coagulation cascade. These observations are now proposed in an organic way through the presentation of 6 possible causal scenarios, presented in the new Figure 5. Among the 6 possible scenarios we also discuss potential collider bias (in response to Reviewer 2, point 4).

Please, find new Table 4, new Figure 5, new Supplementary Tables 8, 9, 10, and 11, and new text in the Methods, where we introduce the eQTL and pQTL lookups (page 7, lines 210-217):

“We queried variants for which we observed potentially mediated effects in the European ancestry datasets of the GTEx Consortium v10 database (https://gtexportal.org/home/; 1-Mar-2025) across 49 tissues (n=65 to 573 samples per tissue) using GTEx API v2 and in the Human Kidney eQTL data [34] (n=686), to assess association with gene expression (eQTL) at P<5×10-8. To identify protein quantitative trait loci (pQTLs) at P<5×10-8, we interrogated GWAS summary results for 2923 plasma proteins on 54,219 UK Biobank participants [35] and 4502 whole blood proteins from a deCODE genetics study [36] (n=35,559; https://decode.com/summarydata; 1-Mar-2025).”

In the Results, where we discuss findings from the eQTL and pQTL investigations (page 10, lines 308-312):

“At SLC34A1, the partial mediation of aPTT corresponded to a 21% larger effect of the SNP on ln(eGFRcrea). eQTL investigation highlighted association with RGS14 expression across most tissues (Table 4). The involved genetic variant was an eQTL for coagulation factor 12 (F12) in the esophagous mucosa and a pQTL for kininostatin, a fragment of kininogen 1 (KNG1), and coagulation factor 2 (F2), all involved in the coagulation cascade (Table 4).”

and in the Discussion, page 12, lines 367-372:

“Statistical mediation observed at SHROOM3 and SLC34A1 is compatible with different causal scenarios (Fig 5). First, given several established connections between both loci and eGFRcrea, it looks implausible that genes and eGFRcrea may independently affect the mediators, generating apparent mediation (Fig 5, scenario 1). Moreover, given similar allele frequencies between CHRIS and CKDGen, collider bias acting through population selection seems implausible (Fig 2A and 2D).”

and pages 13-14, lines 398-412:

“At SLC34A1, confounding by LD seems the most likely possibility. Both RGS14 [52] and SLC34A1 [53] are involved in phosphate handling that is critically dealt by the kidney, also in response to parathyroid hormone (PTH) regulation, as confirmed by our pQTL lookup that identified association with PTH in the deCODE proteomics data. The association with eGFRcrea [38] likely reflects causal involvement of this locus in kidney function regulation. In addition, the locus was previously associated with aPTT in 9240 European ancestry individuals [39] including participants from the same area where the CHRIS study was sampled. By quantifying the clotting time from the activation of factor XII (F12) until the formation of a fibrin clot, aPTT is a measure of the integrity of the intrinsic and common coagulation pathways [54]. The observed mediation, in which adjustment for aPTT increased the SNP-eGFRcrea association by 21%, is likely related to the proximity between SLC34A1 and the coagulation factor gene F12. F12 initiates the coagulation cascade, which in turn involves kininogen 1 (KNG1) and coagulation factor 2 (F2), both of which were identified in our pQTL lookup. Results are thus compatible with the presence of common haplotypes partially tagging both RGS14/SLC34A1 and F12, inducing partially overlapping effects on eGFRcrea and aPTT.”

3. The PDILT snp is usually associated with UMOD - can this be mentioned clarified

Response: We replaced PDILT with UMOD-PDILT throughout.

4. The SHROOM3 findings related to Magnesium are interesting and perhaps a mechanistic diagram would help here to understand how SHROOM3 might be mediating magnesium regulation. Again for the SHROOM3 snp some QTL or candidate cis-regulatory elements (cCREs) analysis for the SNPs of interest should be investigated.

Response: To respond to this point, we can largely refer to the response to your point #2 above. For the specific results pertaining to the SHROOM3 locus, we now provide eQTL and pQTL evidence in the new Table 4, and discuss the causal scenarios that might have originated the evidence of mediation in the new Figure 5 and related discussion.

In addition to what already reported in previous point 2, which was common to both SHROOM3 and SLC34A1 loci, please, find new text related to SHROOM3 in the Results (page 10, lines 301-307):

“At SHROOM3, the partial mediation of Mg corresponded to about 11% effect attenuation (Table 3). eQTL investigation across all genes located in the SHROOM3 locus did not identify any genome-wide significant association with SHROOM3 expression (Table 4), as it would be expected given the developmental nature of the gene [41, 42]. However, SHROOM3 resulted being the most associated gene in the kidney tissue. The variants involved in the mediation were also associated with cystatin C protein levels and with other kidney-function relevant proteins such as NAR3 and CXCL11 (I-TAC) [43] (Table 4).”

And in the Discussion, pages 12-13, lines 372-397:

“At SHROOM3, apparent mediation due to LD confounding is possible: there are several haplotypes spanning all genes in the locus that were jointly associated with eGFRcrea and Mg [43]. However, it is unclear which of such genes might affect Mg without jointly affecting eGFRcrea. Known biology should also bring to exclude a causal effect of Mg on eGFRcrea. For SHROOM3, the most likely scenario looks like that of SHROOM3 having independent causal effects on eGFRcrea and Mg levels. Genetic variants at SHROOM3 have been associated with CKD [8, 9], reduced eGFRcrea [9], increased albumin-to-creatinine ratio [10], and low serum magnesium levels [11]. SHROOM3 is necessary to maintain the glomerular filtration barrier integrity [41]. Variants in this gene have been shown to be associated with increased risk of CKD through disruption of the transcription factor TCF7L2 in podocyte cells [48]. The nearby CCDC158 is involved in renal proximal tubular endocytosis and is expressed in renal proximal tubular cells and in glomeruli of individuals free from nephrocalcinosis [49]. There is thus support for a causal role of the SHROOM3 locus on kidney function, at least through SHROOM3 itself, but not excluding CCDC158. On the magnesium side, one can note that SHROOM3 is also expressed in the epithelia morphogenesis of the gut [50], which has a crucial role in Mg absorption and regulation. This evidence would be compatible with a previous observation that Mg association with the SHROOM3 locus did not change when adjusting for eGFRcrea, suggesting pleiotropic independent effects [40]. However, being serum magnesium a poor marker of dietary magnesium [51] this hypothesis should be considered with caution. On the other hand, our observation of a partial mediation (Mg adjustment caused 11% reduction of SHROOM3 effect on eGFRcrea), would be compatible with the kidney’s role on Mg regulation, through excretion and reabsorption. By this reasoning, high Mg levels could be partially determined by long-term lower kidney function levels due to SHROOM3 variations affecting embryonal kidney development. Altogether, combining Fig 5’s possible scenario 3 and likely scenario 6 would imply the partial mediation as the result from both a direct effect of SHROOM3 on Mg and an indirect effect of SHROOM3 on Mg through lower kidney function.”

5. It wou

---

## [Decision Letter · Decision Letter 1]

Systematic mediation and interaction analyses of kidney function genetic loci in a general population study

PONE-D-24-44950R1

Dear Dr. Ghasemi-Semeskandeh,

We’re pleased to inform you that your manuscript has been judged scientifically suitable for publication and will be formally accepted for publication once it meets all outstanding technical requirements.

Kind regards,

Andrew J. Mallett

Academic Editor

PLOS ONE

Additional Editor Comments (optional):

Reviewers' comments:

Reviewer's Responses to Questions

**Comments to the Author**

Reviewer #1: All comments have been addressed

Reviewer #2: All comments have been addressed

Reviewer #3: All comments have been addressed

2. Is the manuscript technically sound, and do the data support the conclusions?

Reviewer #1: Yes

Reviewer #2: Yes

Reviewer #3: Yes

3. Has the statistical analysis been performed appropriately and rigorously?

Reviewer #1: Yes

Reviewer #2: Yes

Reviewer #3: Yes

4. Have the authors made all data underlying the findings in their manuscript fully available?

Reviewer #1: Yes

Reviewer #2: Yes

Reviewer #3: (No Response)

5. Is the manuscript presented in an intelligible fashion and written in standard English?

Reviewer #1: Yes

Reviewer #2: Yes

Reviewer #3: Yes

Reviewer #1: The revised manuscript has answered my previous concerns. The updated draft allows a more complete understanding of the data and its interpretation.

Reviewer #2: All of my comments have been thoroughly addressed, and after reviewing the changes, I don’t have any further suggestions.

Reviewer #3: (No Response)

**Do you want your identity to be public for this peer review?** For information about this choice, including consent withdrawal, please see our Privacy Policy

Reviewer #1: No

Reviewer #2: No

Reviewer #3: No

---

## [Editor Report · Acceptance letter]

PONE-D-24-44950R1

PLOS ONE

Dear Dr. Ghasemi-Semeskandeh,

I'm pleased to inform you that your manuscript has been deemed suitable for publication in PLOS ONE. Congratulations! Your manuscript is now being handed over to our production team.

Kind regards,

on behalf of

Professor Andrew J. Mallett

Academic Editor

PLOS ONE